# MatchEx: Model-Level GNN Explanations with Multi-Granular Insights

## Abstract

Graph Neural Networks (GNNs) are increasingly deployed in high-stakes domains where interpretability is crucial. Existing model-level explanation methods largely rely on generative models, which often produce motifs that fail to resemble real instances, cannot account for the diversity of discriminative motifs recognized by the classifier for a target class and lack mechanisms for translating global explanations to instance-level insights. We present MatchEx, a framework that discovers discriminative motifs directly from real instances by optimizing a novel matching objective. Unlike isomorphism, which can only recover identical motifs that rarely occur in real-world graphs, this objective extends beyond exact matches to provably recover semantically similar motifs, allowing generalizable explanations. The matching mechanism also enables projection of class level rationales onto individual graphs for faithful instance-level insights. When a single motif fails to explain all instances, MatchEx adaptively partitions the instances in a class into coherent subgroups with distinct rationales. Extensive experiments across six real and synthetic datasets show that MatchEx consistently outperforms state-of-the-art baselines, delivering coherent, generalizable, and multi-granular explanations.

## 1 Introduction

Graph Neural Networks (GNNs) have achieved state-of-the-art performance across diverse tasks involving graph-structured data. This success has driven their adoption in critical application domains such as drug discovery and synthesis (Merchant et al., 2023), weather forecasting (Lam et al., 2023), and recommender systems (Wu et al., 2022). In high-stakes settings such as predicting polypharmacy side effects (Zitnik et al., 2018) and detecting fake news (Xie et al., 2025), the growing use of GNNs has underscored the need for interpretability and transparency in their decision-making processes. Consequently, recent research has placed significant emphasis on developing post-hoc explainability methods tailored for GNNs. According to a recent survey (Kakkad et al., 2023), post-hoc explainability techniques for GNNs can be broadly categorized into two groups: *instance-level* and *model-level* methods. Instance-level explainers (Vu & Thai, 2020; Ying et al., 2019) aim to highlight a salient subgraph within a specific graph instance that influences the model's prediction the most. However, such explanations are localized and typically do not generalize to other instances, limiting their effectiveness in revealing the model's overall decision strategy. In contrast, model-level explainers (Yuan et al., 2020; Wang & Shen, 2023) seek to identify class-discriminative motifs that are consistently used by the model to identify instances of a specific class. These explanations are more generalizable, require less human scrutiny, and provide insights into the model's reasoning that hold across a broad range of instances.

However, existing model-level approaches (Wang & Shen, 2023; Chen et al., 2024; Yuan et al., 2020; Wang & Shen, 2024; Saha & Bandyopadhyay) predominantly rely on graph generative models to synthesize such motifs. This reliance introduces key limitations. Most notably, the generated explanations often fail to resemble actual motifs present in real instances of the target class. To ensure that synthesized explanations remain consistent with the underlying data domain, these models often require domain-specific constraints which are difficult to design without expert knowledge. Additionally, the design of the generative model itself also limits the ability to produce graphs with varied node and edge features. On the other hand, discovery based approaches such as PAGE (Shin et al., 2024) which search for a common discriminative subgraph within graphs classified to the tar-

get class, struggle to converge as the search space grows exponentially with graph size and number of instances. Additionally both generative and discovery based approaches exhibit low local fidelity. They lack a mechanism to connect the model-level rationale to instances classified to the class. This makes it unclear how the synthesized explanation is tied to the classifer behavior on the instances within the class. When a target class comprises subgroups of graphs that the classifier distinguishes using distinct motifs, these approaches are prone to mode collapse, capturing only one motif and ignoring others. This significantly limits their utility in high-stakes domains, such as drug toxicity prediction, where capturing the full diversity of discriminative motifs is essential.

In light of the limitations of existing model-level approaches, we propose **MatchEx**, a framework that discovers model-level explanations from instances classified to the target class rather than synthesizing them using a generative model. Instead of searching for a common discriminative motif across instances which becomes computationally infeasible as graph size and dataset scale increase, MatchEx operates by optimizing a novel matching objective guided by a graph matching algorithm to discover a shared class level rationale. It does so by aligning semantically similar motifs across instances that need not be strictly isomorphic but instead share equivalent structural roles or functional patterns, unlike approaches that attempt to find a single common subgraph that rarely exists in real-world data. The same matching mechanism also projects the class-level rationale back onto individual instances, establishing an explicit bridge between global and local behavior that prior approaches lacked. It is further used to devise a generalization score to detect when a single explanation does not sufficiently generalize across all instances of the class. In such cases, MatchEx can identify subgroups of instances within the class that share a common rationale. Experiments on six diverse real and synthetic datasets, including the large-scale OGB-Molhiv benchmark, show that MatchEx outperforms state-of-the-art model-level explainers. In addition, it reveals what the model has learned, diagnoses biases and pitfalls, and offers multi-granular explanations across global, subgroup, and instance levels, underscoring its effectiveness and broad applicability.

## 2 MATCHEX

The objective of MatchEx is to identify a common discriminative motif shared across a broad set of instances within a target class that the classifier consistently relied on for assigning the target class label. To fulfill this goal, the same motif must serve as the decisive factor in determining the class identity for all relevant instances. However, expecting the presence of isomorphic motifs across all such instances is unrealistic in most real-world scenarios. Rather, what is typically observed are semantically similar motifs that capture the same overall structural pattern, albeit with variations in the number or arrangement of constituent nodes. Therefore, the task of a model-level explainer is to discover a motif that is semantically aligned with the class-discriminative structures present in the instances of the target class. To support this, we incorporate supervision via a multi-graph matching algorithm that promotes the discovery of a motif that semantically resembles the discriminative substructures across instances.

### 2.1 MULTI-GRAPH MATCHING AND CLUSTERING PROBLEM

We use the graduate assignment approach (Wang et al., 2020a) to solve a graph matching problem on the instances that belong to a target class. Let $D_c$ denote the set of graphs that have been classified to the target class $c$ by the classifier. Each graph $G_i \in D_c$ can be represented as a tuple $G_i = (X_i, A_i)$, where $X_i \in \mathbb{R}^{n_i \times l}$ denotes the $l$-dimensional feature vectors for the $n_i$ nodes. The matrix $A_i \in \mathbb{R}^{n_i \times n_i}$ is a weighted adjacency matrix capturing the connectivity within $G_i$.

The objective of the graduate assignment algorithm is to return a node matching matrix $M_{ij} \in \{0,1\}^{n_i \times n_j}$ for any two graphs $G_i, G_j \in D_c$, which defines the correspondence between their nodes. Note that the multi-graph matching (MGM) problem differs significantly from a pairwise matching problem. Here, it is crucial to enforce the principle of **cycle-consistency**. Cycle consistency enforces the transitive consistency of matching across multiple graphs. Specifically, if node $a$ in graph $G_i$ is matched to node $b$ in graph $G_j$, and node $b$ is matched to node $c$ in graph $G_k$, then node $a$ should also be matched to node $c$ via transitive composition. Formally, the set of pairwise matchings $\{M_{ij}\}$ is said to be cycle-consistent if $\forall i, j, k \in \{1, \ldots, |D_c|\}$, $M_{ij} = M_{ik}M_{kj}$.

The graduate assignment algorithm (Wang et al., 2020a) enforces this condition. This condition allows decomposition of all pairwise matchings into alignments between each graph and a shared node universe of size $d$. Let $U_i \in \{0, 1\}^{n_i \times d}$ denote the matching from $G_i$ to this shared universe. Then, the pairwise matching from $G_i$ to $G_j$ can be derived as $M_{ij} = U_i U_j^\top$. Hence, the matching from a graph $G_i$ to the node universe can be transferred to the matching from $G_i$ to another graph $G_j$. As we demonstrate later, we utilize this mechanism to transfer explanations obtained on one graph in a target class to find matching explanations in other graphs belonging to the same class.

It is important to note here that the graduate assignment algorithm solves a more general multi graph matching and clustering(MGMC) problem. Formally, the objective formulated by a slight modification of the Koopmans-Beckmann Quadratic Assignment Problem (KB-QAP) is written as:

$$\max_{\{M_{ij}\},\, i,j \in [|D_c|]} \sum_{i,j \in [|D_c|]} C_{ij} \left( \lambda \operatorname{tr}(M_{ij}^\top A_i M_{ij} A_j) + \operatorname{tr}(M_{ij}^\top W_{ij}) \right) \tag{1}$$

The optimization aims to jointly maximize structural and node-level agreement between graphs that belong to the same cluster. Here, $C_{ij}$ indicates whether graphs $G_i$ and $G_j$ are assigned to the same cluster ($C_{ij} = 1$) or not ($C_{ij} = 0$). The term $\operatorname{tr}(M_{ij}^\top A_i M_{ij} A_j)$ measures the structural consistency between the adjacency matrices $A_i$ and $A_j$ under the assignment $M_{ij}$, while $W_{ij} = X_i X_j^\top$ captures the similarity at the node level based on the correspondence of features. When the task is solely multi-graph matching without clustering, the cluster membership is implicitly assumed to be fixed with $C_{ij} = 1$ for all pairs $i, j$ such that all graphs belong to a single cluster. The MGMC formulation extends the classical notion of graph isomorphism as formalized in the following theorem.

**Theorem 1** (**Matching Generalizes Isomorphism**). *Let $\{G_i = (X_i, A_i)\}_{i=1}^m$ be a collection of graphs and consider the MGMC objective in Eq. 1. For any pair $(i, j)$ with $C_{ij} = 1$, the maximizer $M_{ij}^\star$ recovers (i) the exact permutation when $G_i$ and $G_j$ contain isomorphic induced subgraphs and (ii) the maximum common induced subgraph (MCIS) when no isomorphism exists (for large $\lambda$).*

The proof of Theorem 1 is deferred to Appendix A.

## 2.2 Discovery of Model-Level Explanations using MatchEx

Let $\mathcal{D}_c = \{G_i\}_{i=1}^{N_c}$ denote the set instances classified to class $c$ by the trained classifier $f : G \to Y$. We start out by computing the multi-graph matching by optimizing Equation 1 with $C_{ij=1}$ for all instances in $\mathcal{D}_c$. Our objective is to learn shared motifs $\mathcal{M} = \{M_1, \cdots, M_k\}$ that explain the class identity of all instances in $D_c$. The primary goal is that each motif $M_i$ should explain the largest possible subset of instances in $\mathcal{D}_c$. To assess this, we later devise a generalization score that evaluates the range of instances explained by a motif $M_i$. For now, note that any discovered motif $M_i$ is a subgraph of some graph $G_i \in \mathcal{D}_c$. Therefore, we may use the matching assignment matrices to obtain a subgraph $M_j$ in any $G_j \in \mathcal{D}_c$ that matches $M_i$. This allows us to verify whether the matched subgraph $M_j$ also explains the class identity of $G_j$.

### 2.2.1 Design and Learning Objective of the Explainer

To identify representative explanations for a target class, we focus on the set of graphs that are confidently classified into that class by the trained GNN classifier. Since the motifs that the model truly relies on to identify a target class are unobservable, the standard assumption in model-level literature is that motifs with high target class scores when evaluated by the model $f$ contain discriminative information. Following this principle, we posit that instances in $\mathcal{D}_c$ receiving high class scores are most likely to contain discriminative substructures used by the model to identify the target class $c$.

We rank the graphs in a class according to their predicted class scores. We pose the explainer the task of finding one motif each from the top $k$ graphs. The objective of the explainer is designed with inspiration from the information bottleneck principle such that it optimizes a combination of two opposing objectives: a) the target class score of the obtained motif in a graph $G_i$ and the target class scores of the matched subgraphs to the motif in the other $k - 1$ graphs to the obtained explanation is high b) the size of the obtained motif in each graph is small. The explainer is parametrized as a GNN followed by a linear layer and is trained to maximize this objective. For a graph $G_i$ in the top $k$ graphs containing $n_i$ nodes, the explainer outputs a soft node selection mask $\mathbf{m}_i \in [0, 1]^{n_i}$.

To make node selection differentiable, we adopt the Gumbel-Sigmoid reparameterization trick. The explainer outputs a set of logits $\mathbf{z}_i \in \mathbb{R}^{n_i}$, from which the stochastic mask $\mathbf{m}_i$ is sampled as:

$$\mathbf{m}_i = \sigma \left( \frac{\log \mathbf{u} - \log(1 - \mathbf{u}) + \mathbf{z}_i}{\tau} \right), \quad \mathbf{u} \sim \text{Uniform}(0, 1) \tag{2}$$

where $\sigma(\cdot)$ denotes the sigmoid function and $\tau$ is the temperature parameter.

The masked node features are obtained as $X_i' = \mathbf{m}_i \odot X_i$, and a soft edge mask $\mathbf{e}_i$ is implicitly induced by computing $e_{uv} = m_{iu} \cdot m_{iv}$ for each edge $(u, v)$. The mask weighted adjacency matrix can be obtained as $A_i' = \mathbf{e}_i \odot A_i$. Note that the classifier is kept frozen during explainer training. The class score objective can be defined as $\mathcal{L}_{\text{cls}} = f_c(X_i', A_i')$ where $f_c(.)$ depicts the class score corresponding to the target class $c$.

To encourage generalizable explanations, an alignment objective is introduced which is computed across the top $k$ graphs. Recall that, $\mathbf{U}_i \in \{0, 1\}^{n_i \times d}$ denotes the node matching matrix from $\mathcal{G}_i$ to a universal node space of dimension $d$. The explanation mask $m_i$ on a graph $G_i$ can be transferred to another graph $G_j$ using $\mathbf{m}_j^{trans} = \mathbf{U}_j \mathbf{U}_i^\top \mathbf{m}_i$. The masked node feature matrix $X_j^{trans'}$ and adjacency matrix $A_j^{trans'}$ can be calculated in the same manner using the transferred mask. The alignment objective encourages the explainer to find an explanation such that these transferred masks also determine the class identity of the other graphs $\mathcal{L}_{\text{matching}} = \sum_{j \neq i} f_c(X_j^{trans'}, A_j^{trans'})$.

The bottleneck on the size of the explanation is imposed using a $L_1$ norm regularization $\|m_i\|_1$ on the mask values and optionally a budget constraint is used when needed $\mathcal{L}_{\text{budget}} = \text{ReLU} \left( \sum_{v=1}^{N_i} m_{i,v} - B \right)$.

The final training objective combines all components as:

$$\mathcal{L}_{\text{MatchEx}} = \mathcal{L}_{\text{cls}} + \lambda_{\text{matching}} \mathcal{L}_{\text{matching}} + \lambda_{\text{sparsity}} \|m_i\|_1 + \lambda_{\text{budget}} \mathcal{L}_{\text{budget}} \tag{3}$$

In addition, we introduce two lightweight regularizers to keep the explanations human interpretable: a connectivity term, which encourages neighboring nodes to receive similar mask values, and an entropy term, which avoids ambiguous values and pushes them towards $0, 1$. They are discussed in detail in Appendix E.

### 2.3 Subgroup Discovery and Selection of the Final Model-Level Explanation

Optimization of $\mathcal{L}_{\text{MatchEx}}$ yields $k$ candidate model-level explanations, each derived from one of the top-$k$ graphs ranked by their predicted class scores. To evaluate how well a candidate explanation generalizes across the class, we define the generalization score of a motif $M$ as

$$g_c(M) = \frac{1}{|\mathcal{D}_c|} \sum_{G_j \in \mathcal{D}_c} \mathbb{I}\left[ \left| f_c(\text{Match}(M, G_j)) - f_c(G_j) \right| \leq \gamma \right], \tag{4}$$

where $f_c(G_j)$ is the class score of graph $G_j$ for class $c$, $\text{Match}(M, G_j)$ denotes the subgraph of $G_j$ matched to motif $M$, and $\gamma$ is set to 0.1 in our experiments. Intuitively, $g_c(M)$ measures the fraction of graphs whose class scores remain consistent when explained through the matched motif. The candidate motif with the highest $g_c(M)$ is selected as the representative model-level explanation for the class.

If the $g_c$ for all $k$ candidates fall below a tolerance threshold (set to 0.7 in our experiments), MatchEx infers that no single explanation sufficiently generalizes across the class. In this case, the multi-graph matching and clustering (MGMC) procedure is applied to partition the graphs into more coherent clusters. MatchEx is then invoked within each cluster to identify subgroup-level explanations. When some clusters still fail to yield a satisfactory explanation, MGMC is recursively applied within those clusters, thereby refining the partition hierarchically. This ensures adaptivity while avoiding repeated global recomputation. Algorithm 2 illustrates the pseudocode for MatchEx.

**Recovering Matching Instance-Level Explanations.** The matching assignment matrices also enable recovery of instance-level explanations from a model- or subgroup-level explanation. Let the

final model-level explanation mask be $m_i^*$ corresponding to a graph $G_i^*$ among the top $k$ candidates. A transferred explanation on another graph $G_j$ in the same subgroup can be obtained using:

$$m_j = U_j U_i^{*\top} m_i^*, \tag{5}$$

where $U_i^*$ and $U_j$ are the assignment matrices for $G_i^*$ and $G_j$, respectively. The node mask $m_j$ allows sampling of an instance-level explanation on $G_j$ with induced edge weights. This natural scheme equips MatchEx with greater local fidelity, enabling users to interpret explanations at finer granularity and establishes a direct link between global and local explanations.

## 3 EXPERIMENTAL EVALUATION

We evaluate MatchEx across six datasets of varying scales, including both real-world and synthetic benchmarks, to assess its ability to discover model-level explanations. For each dataset, we first train a classifier and collect the set of instances $\mathcal{D}_c$ that the classifier assigns to a target class $c$. MatchEx is then applied to $\mathcal{D}_c$ with the objective of identifying motifs that serve as shared explanatory rationales for instances in the class. We make the code available at https://anonymous.4open.science/r/MatchEx-D6A4/.

We evaluate MatchEx through three research questions: **R1:** Can it discover a motif that explains the largest possible subset of instances in $\mathcal{D}_c$? **R2:** When a single explanation is insufficient, can it identify coherent subgroups within $\mathcal{D}_c$ and extract subgroup-specific motifs? **R3:** Can instance-level explanations for specific graphs be faithfully recovered from the discovered model-level or subgroup-level motifs? Before addressing these questions, we first introduce the evaluation metrics used to assess the performance of MatchEx.

### 3.1 METRICS FOR EVALUATION

We employ a range of evaluation metrics to rigorously assess different aspects of the explanatory capability of MatchEx.

**Model-Level Metrics.** Since there is no ground truth available for the motifs used by the classifier to recognize a target class, the evaluation of discovered explanations must rely on proxy metrics. We adopt three key metrics to assess the quality of model-level explanations: **Target Class Score**, **Generalization Score**, and **Wasserstein Distance**.

**Target Class Score ($p_c$).** For a discovered explanatory motif $M$, the target class score $p_c$ denotes the classifier's predicted probability that $M$ belongs to class $c$. A higher score indicates that the classifier confidently associates the motif with the target class. This metric has been widely adopted in prior model-level explanation literature (Yuan et al., 2020; Wang & Shen, 2023; Chen et al., 2024; Saha & Bandyopadhyay). We report the mean and standard deviation of $p_c$ across 50 explanations on each target class.

**Generalization Score ($g_c$).** As described by Equation 4, it evaluates the fraction of graphs in $\mathcal{D}_c$ whose own class scores remain within a $\gamma$ threshold of the subgraph which is matched to the motif $M$. We set $\gamma$ to 0.1. Note that the generalization score cannot be computed for generative model based explainers such as XGNN Yuan et al. (2020) and GNNInterpreter Wang & Shen (2023), since there is no guarantee that the generated motif would be part of any graph $G_i \in \mathcal{D}_c$. We report the mean and standard deviation of $g_c$ across 50 explanations on each target class.

**Wasserstein Distance ($W_1$).** While $p_c$ and $g_c$ assess the classifier's confidence and motif transferability, they do not capture how closely the distribution of discovered motifs aligns with the distribution of true class instances. To address this, we compute the Wasserstein-1 (Earth Mover's) distance between the embeddings of the discovered motifs and those of graphs in $\mathcal{D}_c$. Since the number of explanation motifs may be small compared to the number of instances in the class, we expand this set by resampling each motif embedding with zero-mean Gaussian noise, where the variance is set to a small fraction (10%) of the average pairwise distance between motifs. This ensures perturbations remain centered on each motif while producing a denser support for stable estimation. A smaller $W_1$ indicates that the motifs lie closer to the support of the true class distribution, thereby reflecting better semantic alignment. Unlike the generalization score, $W_1$ can be computed for both retrieved and generative explanations, enabling a fair comparison across methods.

**Instance-Level Metrics.** For completeness, we also evaluate matching instance-level explanations produced by MatchEx using **Fidelity**, **Infidelity**, and **Sparsity**, which measure predictive consistency, necessity of the explanation, and conciseness of the selected subgraph. Full definitions and results are provided in Appendix B.

## 3.2 EXPERIMENTS AND RESULTS

Our experiments span four real-world datasets (IMDB-Multi (Yanardag & Vishwanathan, 2015), REDDIT-Binary (Yanardag & Vishwanathan, 2015), MUTAG (Debnath et al., 1991), OGB-MOLHIV (Hu et al., 2020)) and two synthetic datasets (BA-2Motif (Luo et al., 2024), Group-Shapes), with full details provided in Appendix C. Among the synthetic benchmarks, the Group-Shapes dataset is particularly important as it consists of two classes where instances can differ substantially in structure: one class contains Star and Lollipop graphs, while the other includes Grid and Tree graphs. This setup reflects scenarios where a single class does not admit a uniform explanation, requiring the discovery of meaningful subgroups and their associated motifs. To evaluate scalability, we further include the OGB-MOLHIV dataset, which comprises 41,127 molecular graphs labeled for HIV activity prediction. We compare MatchEx against three representative model-level explainers: XGNN (Yuan et al., 2020), a reinforcement learning-based approach that constructs motifs node by node, GNNInterpreter (Wang & Shen, 2023), which employs a probabilistic generative model to synthesize explanations and PAGE (Shin et al., 2024) which is a discovery-based approach that searches for common discriminative subgraphs across instances classified to the target class. This selection provides a balanced comparison across both generative and discovery based paradigms.

Figure 1: Model-Level Explanations across Target Classes.

**R1: Capability to generate a common model-level explanation** We first evaluate the ability of MatchEx to discover motifs that serve as common rationales for all instances in a target class. Except on the OGB-MOLHIV and GroupShapes datasets, MatchEx considered instances in all target classes

of other datasets as a single group, yielding a single common explanation for each class. As shown in Table 1, MatchEx achieves higher target class scores and generalization scores on almost all datasets, indicating that its retrieved motifs are strongly associated with the classifier's notion of the target class and matching discriminative motifs recur across many confidently classified instances, highlighting their role as class-level rationales. Lower $W_1$ scores for MatchEx also indicate that the discovered explanations faithfully capture the distribution of the target class learnt by the classifier. Figure 1 further illustrates representative motifs obtained on each dataset.

Across datasets, MatchEx consistently recovers class-defining motifs. On **MUTAG**, it identifies fused ring structures and $NO_2$ as the rationale for mutagenic compounds, while for non-mutagenic ones the explanation is dominated by halogen atoms, exposing a spurious association learnt by the classifier confirmed by an analysis in Appendix F. On **IMDB-Multi**, Action(Class 0) graphs show densely interconnected high-degree actors, whereas Comedy and Drama( Class 1 & 2) graphs feature one or two central actors with peripheral groups, aligning with strong generalization scores in Table 1. On **REDDIT-B**, hub-and-spoke patterns emerge for the Question–Answer(Class 0) class and clustered interactions for the Discussion class (Class 1), reflecting the communication dynamics. learnt by the classifier. Finally, on **BA-2Motif**, MatchEx retrieves the planted House and Cycle motifs, validating its ability to recover ground-truth structures in synthetic benchmarks.

The baselines show clear limitations compared to MatchEx: XGNN performs well on MUTAG but degenerates to trivial explanations such as line graphs or isolated nodes on other datasets; GN-NInterpreter produces reasonable motifs on IMDB-Multi and REDDIT-B but produces pathological unrealistic motifs on MUTAG. While PAGE discovers motifs from real graphs, it often does not manage to find motifs with optimal class scores or high generalization scores on large graphs such as REDDIT(Class 1) and the IMDB-Multi dataset with matching but non-isomorphic patterns.

Table 1: Comparative Results for Model-Level Explanations

| Classes (Dataset) | XGNN | | GNNInterpreter | | PAGE | | | MatchEx (Ours) | | |
|---|---|---|---|---|---|---|---|---|---|---|
| | $p_c$ (↑) | $W_1$ (↓) | $p_c$ (↑) | $W_1$ (↓) | $p_c$ (↑) | $W_1$ (↓) | $g_c$(↑) | $p_c$ (↑) | $W_1$ (↓) | $g_c$(↑) |
| Mutagenic (MUTAG) | 0.961±0.004 | 1.986 | 0.981±0.000 | 3.211 | 0.983±0.003 | 1.871 | 0.825±0.010 | **0.998±0.003** | 0.801 | **0.901±0.001** |
| Non-Mutagenic (MUTAG) | **1.000±0.000** | 1.591 | **1.000±0.000** | 4.701 | **1.000±0.000** | 1.761 | 0.894±0.009 | **1.000±0.000** | 1.001 | **1.000±0.000** |
| Class 0 (Reddit-B) | 0.004±0.000 | 10.510 | 0.821±0.014 | 1.911 | 0.776±0.051 | 2.677 | 0.739 ±0.003 | **0.975±0.001** | 1.229 | **1.000±0.000** |
| Class 1 (Reddit-B) | 0.329±0.000 | 10.974 | **0.988±0.000** | **0.992** | 0.745±0.038 | 2.189 | 0.661±0.001 | 0.976±0.000 | 1.041 | **1.000±0.000** |
| Class 0 (IMDB-Multi) | 0.399±0.000 | 7.843 | 0.650±0.020 | 5.619 | 0.955±0.000 | 3.982 | 0.821±0.004 | **0.994±0.006** | 2.636 | **0.991±0.000** |
| Class 1 (IMDB-Multi) | 0.332±0.000 | 9.913 | 0.354±0.020 | 6.729 | 0.856±0.005 | 4.813 | 0.760 ±0.010 | **1.000±0.000** | 1.302 | **0.998±0.000** |
| Class 2 (IMDB-Multi) | 0.415±0.000 | 8.900 | 0.711±0.031 | 3.414 | 0.855±0.004 | 2.793 | 0.836±0.031 | **0.972±0.001** | 1.420 | **0.956±0.000** |
| Cycle (BA-2Motif) | 0.5020±0.003 | 3.202 | 0.947±0.000 | 1.110 | **1.000±0.000** | **0.584** | **1.000±0.000** | 0.999±0.000 | 0.772 | **1.000±0.000** |
| House (BA-2Motif) | 0.491±0.248 | 2.190 | 0.981±0.000 | 1.494 | **1.000±0.000** | 0.867 | **1.000±0.000** | **1.000±0.000** | 0.755 | **1.000±0.000** |
| Star-Lollipop (GroupShapes) | 0.311±0.002 | 4.449 | 0.952±0.000 | 2.758 | 0.998±0.000 | 2.648 | 0.512±0.005 | **1.000±0.000** | 0.998 | **1.000±0.000** |
| Grid-Tree (GroupShapes) | 0.459±0.001 | 3.537 | 0.986±0.000 | 2.511 | 0.975±0.003 | 1.173 | 0.554±0.008 | **0.999±0.007** | 0.303 | **1.000±0.000** |
| Non-HIV (OGBMolhiv) | NA | NA | 0.787±0.001 | 12.967 | 0.511 ±0.058 | 13.300 | 0.337 ± 0.072 | **0.962 ± 0.005** | 2.644 | **1.000±0.000** |
| HIV (OGBMolhiv) | NA | NA | 0.199 ±0.018 | 4.375 | 0.089 ± 0.096 | 4.962 | 0.000 ±0.000 | **0.991 ± 0.007** | 0.982 | 0.004±0.000 |

**R2: Recognizing Subgroups of Instances in a Target Class that share a Common Explanation**
In absence of a common explanation on a target class, MatchEx discovers subgroups for which a common rationale can be found. Wherever subgroups are discovered, Table 1 reports mean $p_c$ and $g_c$ scores across subgroups on those classes.

This behavior is clearly illustrated on the **GroupShapes** dataset. In the Star–Lollipop class, MatchEx discovers two distinct subgroups, one explained by a Star motif and the other by a Lollipop motif. Similarly, in the Grid–Tree class, MatchEx identifies two subgroups corresponding to the Grid and Tree motifs. Competing baselines, by contrast, always generate a non-representative single explanation for the entire class, thereby failing to represent the underlying heterogeneity.

On the large-scale **OGB-Molhiv** dataset (Hu et al., 2020), where the majority class exhibits substantial structural diversity, MatchEx partitions the class into four subgroups, each characterized by a distinct motif depicting distinct functionalities. These subgroup-level explanations achieve notably higher $p_c$ and $g_c$ scores, and markedly lower $W_1$ distances than both GNNInterpreter and PAGE, demonstrating MatchEx's ability to uncover multiple complementary rationales in complex real-world settings. The identified subgroups and their motifs are provided in Appendix H.1. XGNN is omitted on this dataset as it only applies to graphs with discrete node features. For the minority class, none of the baselines produce explanations with high class scores, while MatchEx retrieves high-scoring instances that nonetheless yield low $g_c$ values. Further node deletion experiments in Appendix H.1 show that removing even a single node from a minority-class graph can flip its predicted label. This reveals that the classifier, affected by extreme class imbalance (only ∼4% minority samples), memorizes minority instances rather than learning a generalizable pattern. Together, MatchEx and the generalization score serve as effective diagnostics, exposing such biases in model behavior.

**R3: Recovering Instance-Level Explanations from Model-Level Explanations** For completeness, we evaluate the quality of instance-level explanations recovered from the model-level motifs using the mask transfer scheme described in Section 2.3. Table 2 reports comparative results against two widely used baselines, GNNExplainer (Ying et al., 2019) and PGExplainer (Luo et al., 2020).

Table 2: Comparative Results for Instance-Level Explanations

| Dataset | MatchEx | | | GNNExplainer | | | PGExplainer | | |
|---|---|---|---|---|---|---|---|---|---|
| | Fidelity(↑) | Infidelity(↓) | Sparsity(↑) | Fidelity(↑) | Infidelity(↓) | Sparsity(↑) | Fidelity((↑)) | Infidelity (↓) | Sparsity(↑) |
| **MUTAG** | 0.931±0.095 | 0.068±0.095 | 0.759±0.033 | **0.984±0.125** | **0.015±0.125** | 0.445±0.093 | 0.813±0.290 | 0.186±0.290 | 0.765±0.098 |
| **Reddit-B** | **0.792±0.092** | **0.207±0.092** | **0.841±0.059** | 0.513±0.282 | 0.486±0.282 | 0.558±0.143 | 0.468±0.195 | 0.531±0.195 | 0.681±0.028 |
| **IMDB-Multi** | 0.794±0.049 | 0.205±0.049 | 0.448±0.044 | 0.838±0.223 | 0.161±0.223 | 0.531±0.105 | 0.788±0.151 | 0.211±0.151 | 0.422±0.119 |
| **BA-2Motif** | **0.982±0.014** | **0.017±0.036** | 0.812±0.054 | 0.937±0.005 | 0.062±0.005 | 0.779±0.076 | 0.961±0.024 | 0.038±0.024 | **0.851±0.074** |
| **GroupShapes** | **0.931±0.008** | **0.068±0.008** | **0.512±0.085** | 0.920±0.014 | 0.079±0.014 | 0.416±0.080 | 0.912±0.007 | 0.087±0.007 | 0.455±0.018 |
| **OGB-Molhiv** | **0.866±0.112** | **0.133±0.112** | **0.771±0.032** | 0.544±0.221 | 0.455±0.221 | 0.349±0.112 | 0.632±0.242 | 0.367±0.242 | 0.511±0.245 |

Runtime comparisons on all datasets and a time complexity analysis and hyperparameter settings for MatchEx can be found in Appendix I and E. Taken together the findings in this Section demonstrate that MatchEx uniquely offers coherent explanations across model, subgroup, and instance-level granularities.

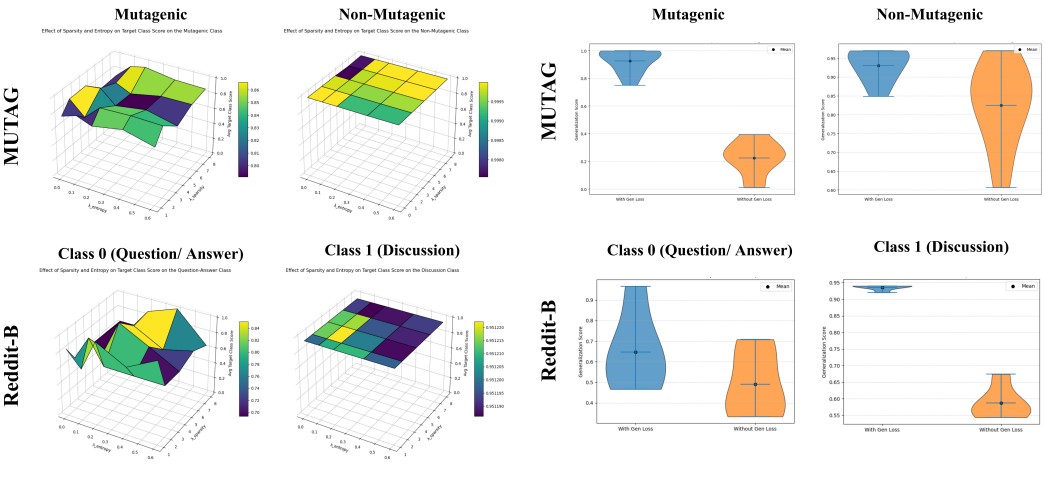

a) Variation in Class-Score with varying Entropy and Sparsity Weights

b) Distribution of Generalization Scores for Explanations with and without the Matching Objective

Figure 2: a) Class Score with variation in $\lambda_{\text{entropy}}$ and $\lambda_{\text{sparsity}}$ b) Generalization Score Distribution of Explanations with (**blue**) and without (**orange**) the matching objective

### 3.3 IMPACT OF DIFFERENT LOSS OBJECTIVES

We assess the effect of the matching objective and regularization terms on the quality of explanations. MUTAG and Reddit-Binary were chosen due to their contrasting scales and domains, providing a representative setting for ablation. As shown in Figure 2b, including the matching loss yields consistently higher generalization scores, whereas removing it produces unstable or weakly transferable explanations. We also vary the regularization weights $\lambda_{\text{entropy}}$ and $\lambda_{\text{sparsity}}$ (Figure 2a), and observe that target class scores remain high throughout indicating high robustness of MatchEx.

## 4 RELATED WORK

**Explanations for GNNs.** Explainability approaches for GNNs can be broadly grouped into two categories: *self-interpretable architectures* and *post-hoc methods*. Self-interpretable GNNs (Miao et al., 2022; Müller et al., 2024) are designed to output not only predictions but also built-in rationales. While this promotes transparency, such approaches typically impose restrictive architectural assumptions that can hinder predictive performance. Post-hoc explainers, to which our work belongs, instead aim to provide explanations for any pre-trained GNN without requiring architectural changes. According to a recent survey (Kakkad et al., 2023), post-hoc methods can be further divided into *instance-level* and *model-level* methods. Instance-level explainers (Pope et al., 2019; Feng et al., 2023; Baldassarre & Azizpour, 2019; Huang et al., 2022; Schlichtkrull et al., 2020; Yuan et al., 2021; Lucic et al., 2022; Lin et al., 2022; Zhang et al., 2021) identify subgraphs, nodes, or edges most influential for a specific input. These provide fine-grained, sample-specific rationales but do not generalize across instances. Hence, inferring general classical behaviour from these require significant human oversight.

Model-level explanations, the subcategory to which MatchEx belongs, aim to uncover discriminative motifs that a classifier relies on to recognize a class. Ideally, these motifs should be explicitly related to instances that the classifier assigns to the class. However, existing model-level methods do not make this connection explicit. Most of them such as XGNN (Yuan et al., 2020), GNNInterpreter (Wang & Shen, 2023), D4Explainer Chen et al. (2024) and Gen-GraphEx Saha et al. (2025) adopt generative approaches which produce explanations that may stray far away from the data distribution. Another line of work is based on discovering model-level explanations, such as PAGE (Shin et al., 2024) which adopts a common subgraph search and motif-scoring procedure, MAGE (Yu & Gao, 2025) and GLGExplainer Azzolin et al. (2023) which build a vocabulary of motifs from which the model-level explanation is built. While this grounds motifs in data, searching or building a vocabulary is computationally expensive, requires prior knowledge of motif size and offers no guarantee of generalization beyond the subset considered. Moreover, in realistic settings isomorphic motifs are unlikely to appear across a large number of instances; one can only expect semantic similarity across explanations, a challenge that MatchEx is explicitly designed to address.

**Multi-Graph Matching and Clustering (MGMC).** MGMC aims to align multiple graphs by establishing consistent node correspondences, thereby revealing shared substructures across instances. Unlike strict subgraph isomorphism, matching does not require graphs to be of the same size, making it better suited for real-world data. Among classical algorithms, Graduated Assignment (Wang et al., 2020a) is attractive because it avoids reliance on anchors (Solé-Ribalta & Serratosa, 2013), does not require initialization (Bernard et al., 2019), and can jointly handle matching and clustering (Wang et al., 2020b). While deep learning-based graph matching methods (Yu et al., 2019; Fey et al., 2020; Nowak et al., 2018) exist, they often depend on costly supervision. MatchEx to the best of our knowledge, is the first GNN explainer to incorporate MGMC for discovering explanations.

## 5 CONCLUSION

In this work, we introduced MatchEx, a novel framework for generating model-level explanations for GNNs. It advances beyond existing approaches by extracting semantically similar motifs rather than relying on strict isomorphism, uncovering subgroups within a class that share a common rationale, and providing an explicit mechanism to connect global motifs to explanations at finer granularities. Together, these contributions position MatchEx as a robust and versatile method for GNN interpretability supporting its deployment in high stakes domains in real world tasks.

**Reproducibility Statement:** We make the code for MatchEx anonymously available in Section 3. Details of datasets, classifier and explainer architecture, hyperparameters and theorem proofs are provided in Appendix C, D, E and A respectively.

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

## A  PROOF OF THEOREM 1

*Proof.* Since all $C_{ij}$ is set to 1. We analyze the optimization objective

$$\Phi(M) \;=\; \lambda \operatorname{tr}(M^\top A_i M A_j) \;+\; \operatorname{tr}(M^\top W_{ij}),$$

where $M$ is a (partial) permutation matrix aligning nodes of $G_i$ and $G_j$, the first term measures structural consistency of the alignment, and the second term measures node feature similarity.

**(a) Isomorphic case.** Suppose $G_i$ and $G_j$ contain induced-isomorphic subgraphs $H_i \subseteq G_i$ and $H_j \subseteq G_j$, related by a bijection $\pi : V(H_i) \to V(H_j)$. Let $P$ denote the permutation matrix corresponding to $\pi$. Because the subgraphs are isomorphic, the adjacency matrices align exactly, i.e. $P^\top A_i P = A_j$ on the motif nodes. Hence the structural term achieves

$$\lambda \operatorname{tr}(P^\top A_i P A_j) \;=\; \lambda\,|E(H_i)|,$$

which is maximal for this subgraph.

Note that corresponding nodes in $H_i$ and $H_j$ also have the same features, so the feature term $\operatorname{tr}(P^\top W_{ij})$ is simultaneously maximized. Any alternative alignment $M \neq P$ must either (i) destroy some edges within $H_i$, lowering the structural term, or (ii) mismatch node features, lowering the feature term. In either case, $\Phi(M) < \Phi(P)$. Thus, the optimal solution coincides with $P$ on the motif nodes (up to automorphisms that yield the same score).

**(b) Non-isomorphic case.** Now suppose $G_i$ and $G_j$ are not isomorphic. Let $S$ denote a maximum common induced subgraph (MCIS) between them, and let $M_S$ denote the alignment that realizes this MCIS. By maximality of $S$, any other alignment $M \neq M_S$ necessarily matches fewer edges. Formally, there exists a margin $\delta_{\text{edge}} > 0$ such that

$$\operatorname{tr}(M_S^\top A_i M_S A_j) - \operatorname{tr}(M^\top A_i M A_j) \;\geq\; \delta_{\text{edge}}.$$

Next consider the feature term. Because node features are bounded and the set of possible alignments is finite, the largest possible difference in feature score between any $M$ and $M_S$ is also bounded:

$$|\mathrm{tr}(M^\top W_{ij}) - \mathrm{tr}(M_S^\top W_{ij})| \;\leq\; \Delta_{\text{feat}} < \infty.$$

If we choose $\lambda$ to be larger than the ratio $\Delta_{\text{feat}}/\delta_{\text{edge}}$, then the structural advantage of $M_S$ outweighs any possible feature advantage of a competing alignment. In particular,

$$\Phi(M_S) - \Phi(M) \;\geq\; \lambda\,\delta_{\text{edge}} - \Delta_{\text{feat}} \;>\; 0,$$

for all $M \neq M_S$. Thus, $M_S$ strictly dominates all alternatives.

**Conclusion.** In the isomorphic case, the optimizer recovers the exact permutation on the motif nodes (up to automorphism). In the non-isomorphic case, for sufficiently large $\lambda$, the optimizer aligns the maximum common induced subgraph. This establishes the claim. $\qquad\square$

## B  Instance-Level Evaluation Metrics

For completeness, we also assess the quality of instance-level explanations produced by MatchEx. We consider three standard metrics: **Fidelity**, **Infidelity**, and **Sparsity**. These capture, respectively, how well the explanation preserves the model's original prediction, how necessary the explanation is for the prediction, and how concise the explanation is in terms of subgraph size.

**Fidelity.** Fidelity measures the drop in the model's confidence for the target class when the explanatory subgraph is removed from the original graph:

$$\text{Fidelity} = \frac{1}{N} \sum_{i=1}^{N} \Big( f_c(G_i) - f_c(G_i \setminus G_{i\mathcal{S}}) \Big),$$

where $f_c(G_i)$ is the prediction score of the full graph, $f_c(G_i \setminus G_{i\mathcal{S}})$ is the score after removing the explanatory subgraph, and $N$ is the number of evaluated instances. A higher fidelity indicates that the detected subgraph is crucial to the class identity of $G_i$.

**Infidelity.** Infidelity measures the drop in the model's prediction score when the complementary subgraph (i.e., everything outside the explanation) is removed:

$$\text{Infidelity} = \frac{1}{N} \sum_{i=1}^{N} \Big( f_c(G_i) - f_c(G_i \setminus G_{i\mathcal{S}}^c) \Big),$$

where $f_c(G_i \setminus G_{i\mathcal{S}}^c)$ denotes the score after masking out the complementary subgraph. A lower infidelity signifies that the explanatory subgraph itself is sufficient for determining the class identity of $G_i$.

**Sparsity.** Sparsity quantifies the fraction of the graph retained in the explanation:

$$\text{Sparsity} = 1 - \frac{1}{N} \sum_{i=1}^{N} \left( \frac{|G_{i\mathcal{S}}|}{|G_i|} \right),$$

where $|G_{i\mathcal{S}}|$ is the number of nodes in the explanatory subgraph and $|G_i|$ is the total number of nodes in the original graph. A higher sparsity corresponds to a more concise and interpretable explanation.

**Reporting.** We report the mean and standard deviation of fidelity, infidelity, and sparsity scores across all target classes on each dataset.

## C  Datasets

The experiments were conducted on fourreal and two synthetic datasets. Table 3 summarises the features of the datasets and the test accuracy of the classifier on each dataset.

Table 3: Dataset Properties and Classifier Accuracy

| Dataset | #Classes | #Graphs | Average #Nodes | Average #Edges | Classifier Accuracy |
|---|---|---|---|---|---|
| **IMDB-Multi** | 3 | 1500 | 19.77 | 96.53 | 0.835 |
| **Reddit-Binary** | 2 | 2000 | 429.63 | 497.75 | 0.812 |
| **MUTAG** | 2 | 188 | 17.93 | 19.79 | 0.8723 |
| **BA-2Motif** | 2 | 1000 | 25 | 25.48 | 1.00 |
| **GroupShapes** | 2 | 1000 | 12.85 | 14.97 | 1.00 |
| **OGB-MOLHIV** | 2 | 41127 | 25.51 | 54.94 | 0.9701 |

### C.0.1 REAL DATASETS

We conduct our experiments on three standard and one large scale graph classification datasets: **IMDB-Multi**(Yanardag & Vishwanathan, 2015), **REDDIT-BINARY (REDDIT-B)**(Yanardag & Vishwanathan, 2015), **MUTAG**(Debnath et al., 1991) and **OGB-MOLHIV**Hu et al. (2020). The IMDB-MULTI Yanardag & Vishwanathan (2015) consists of 1,500 ego-networks extracted from the Internet Movie Database. Each graph corresponds to an actor/actress, with nodes as actors and edges connecting pairs who co-appear in movies. Graphs are labeled into three classes based on the predominant genre of the target actor's movies: Action, Comedy, or Drama. The REDDIT-B dataset comprises graphs of user interactions on Reddit, with nodes denoting users and edges representing reply relationships. Each graph is associated with either a question-answer based or discussion-based community. The MUTAG dataset consists of molecular graphs, where nodes represent atoms and edges denote chemical bonds. Each graph is labeled to indicate whether the compound is mutagenic or non-mutagenic. OGBG-MOLHIV, part of the Open Graph Benchmark (OGB)Hu et al. (2020), contains 41,127 molecular graphs with the binary prediction task of determining whether a molecule inhibits HIV replication. Rich atom-level (type, chirality, valence) and bond-level features are provided. Dataset splits are defined using scaffold splitting, ensuring structurally distinct molecules across train, validation, and test sets. It should be noted that this a highly imbalanced dataset with about $4\%$ of the instances belonging to the minority HIV class, hence classifiers trained on this dataset are often inherently biased towards the majority class.

### C.0.2 SYNTHETIC DATASETS

We also conduct experiments on two synthetic datasets: **BA-2Motif**(Luo et al., 2024) and **GroupShapes** dataset. BA-2Motif consists of Barabasi-Albert graphs with labels assigned by the motif embedded in the graph. Each graph may either contain a House or a Cycle motif.

---

**Algorithm 1** GroupShapes Dataset Generation

---

**Class 0:** {STAR, LOLLIPOP}    **Class 1:** {GRID, TREE}
**Output:** A dataset of graphs labeled by class
**Procedure:**

- For $i = 1$ to $n$:
    - Sample STAR graph $G_{star}$ with $k \in \{5, \ldots, 10\}$ leaves
    - Sample LOLLIPOP graph $G_{lollipop}$ with head $m \in \{5, \ldots, 8\}$ and tail $t \in \{3, \ldots, 5\}$
    - Assign label 0 to both graphs and store $(G_{star}, 0)$, $(G_{lollipop}, 0)$
    - Sample GRID graph $G_{grid}$ of size $p \times q$, $p, q \in \{7, 16\}$
    - Sample TREE graph $G_{tree}$ as binary tree of height $h \in \{3, 7\}$
    - Assign label 1 to both graphs and store $(G_{grid}, 1)$, $(G_{tree}, 1)$

---

The GroupShapes dataset includes two distinct classes of graphs. The first class comprises graphs structured as either a Star or a Lollipop, while the second class contains graphs shaped like a Grid or a Tree. This dataset is motivated by real-world scenarios where instances from the same class can differ significantly in structure. In such cases, it is crucial for the explainer to identify subsets of instances within a class that the classifier recognizes using a shared rationale and to uncover the

common motif underlying this decision. Algorithm 1 demonstrates the pseudocode for generating this dataset.

# D   CLASSIFIER AND EXPLAINER ARCHITECTURE DETAILS

**BA-2Motif.**   The classifier for the BA-2Motif dataset consisted of a two-layer Graph Convolutional Network encoder followed by a linear classification head. Each convolutional layer applied a GCNConv operation with batch normalization, ReLU activation, and dropout with probability 0.5. The hidden dimensionality was set to 64. Node embeddings were aggregated by global mean pooling to obtain a graph representation, which was passed through a fully connected layer for final logits. The model was trained with cross-entropy loss, using the Adam optimizer and early stopping based on validation performance.

**GroupShapes.**   The GroupShapes dataset was modeled with a three-layer Graph Convolutional Network, each layer using the same block structure as in BA2 with hidden dimension 64. Global mean pooling produced graph embeddings, followed by a linear projection to class logits. The deeper architecture was required to capture more complex structural patterns in the dataset. The model was optimized with Adam at learning rate $10^{-3}$ and weight decay $10^{-5}$.

**IMDB-Multi.**   The IMDB-Multi dataset was handled with a two-layer Graph Convolutional Network encoder and a linear classification head. Each layer used GCNConv with ReLU activation and dropout with probability 0.5, with hidden dimension 128. Global mean pooling aggregated node embeddings into graph features, which were mapped to binary logits by a linear readout. The classifier was trained with cross-entropy loss and Adam optimizer, with stronger dropout to mitigate overfitting on large and noisy graphs.

**MUTAG.**   The MUTAG dataset employed a two-layer Graph Convolutional Network with hidden dimension 64. Each layer consisted of GCNConv with ReLU activation and dropout. Embeddings were pooled using global mean pooling, and a linear layer projected the pooled representation into two output logits. Batch normalization was applied after each convolution to stabilize training. Due to the small dataset size, early stopping and moderate weight decay of $10^{-5}$ were used to avoid overfitting.

**Reddit-Binary.**   The Reddit-Binary dataset required a three-layer Graph Convolutional Network with hidden dimension 128 to handle its large and structurally diverse graphs. Each layer followed the standard structure of GCNConv with batch normalization, ReLU activation, and dropout with probability 0.5. Node embeddings were aggregated via global mean pooling, and a linear layer projected the pooled vector to the output space. The model was optimized with cross-entropy loss, Adam optimizer, and a tuned decay schedule to stabilize convergence.

**OGB-Molhiv.**   We employ a Graph Isomorphism Network (GIN) architecture. The encoder consists of five stacked GINConv layers, each parameterized by a two-layer MLP with ReLU activation. Batch normalization and dropout ($p = 0.5$) are applied after each layer to improve stability and prevent overfitting. The final node embeddings are aggregated using global mean pooling to obtain a fixed-size graph-level representation. This pooled representation is passed through a fully connected layer with ReLU activation, followed by a linear classifier that outputs logits over the class labels. The model is trained using the Adam optimizer with a learning rate of $10^{-3}$ and a cross-entropy loss function. A StepLR scheduler with decay factor $\gamma = 0.5$ is applied every 20 epochs to reduce the learning rate adaptively.

**Explainer.**   The explainer architecture consists of a GCN layer that maps input node features to hidden representations, followed by a linear projection that assigns a scalar importance logit to each node. These logits are transformed into differentiable node masks using the Gumbel-sigmoid trick, where annealed temperature scheduling gradually sharpens the masks from soft continuous values toward near-binary selections across training epochs. In this way, the model produces stochastic yet differentiable node-level masks that can be optimized end-to-end to highlight the subgraphs most relevant for explanation.

# E  REGULARIZATION COMPONENTS AND HYPERPARAMETER SETTING

## E.1  ADDITIONAL REGULARIZERS

**Connectivity.**  To discourage scattered explanations, we add a connectivity regularizer that encourages neighboring nodes to receive similar mask values. Let $L_i = D_i - A_i$ denote the graph Laplacian of $G_i$, where $D_i$ is the degree matrix and $A_i$ is the adjacency matrix. The regularizer is defined as

$$\mathcal{L}_{\text{conn}} \;=\; \frac{1}{|E_i|}\,\mathbf{m}_i^\top L_i \mathbf{m}_i \;=\; \frac{1}{|E_i|} \sum_{(u,v)\in E_i} a_{uv}\,(m_{iu} - m_{iv})^2. \tag{6}$$

Minimizing $\mathcal{L}_{\text{conn}}$ promotes smoothness of the mask across edges, leading to more compact and connected explanations instead of disjoint fragments.

**Entropy.**  To improve interpretability, we encourage mask values to approach binary selections. Given $m_{iv} \in (0,1)$ as the soft mask value on node $v$, the entropy penalty is

$$\mathcal{L}_{\text{entropy}} \;=\; -\frac{1}{n_i} \sum_{v=1}^{n_i} \Big[ m_{iv} \log m_{iv} + (1 - m_{iv}) \log(1 - m_{iv}) \Big]. \tag{7}$$

This penalty is minimized when $m_{iv}$ is close to 0 or 1, thereby producing crisp, interpretable masks that are more stable when transferred across graphs.

## E.2  SETTING OF THE EXPLAINER HYPERPARAMETERS

The hyperparameters for training the Explainer on all the datasets are detailed in Table 4.

Table 4: Explainer hyperparameters used for each dataset.

| Dataset | $\lambda_{\text{entropy}}$ | $\lambda_{\text{sparsity}}$ | $\lambda_{\text{budget}}$ | **Budget** | $\lambda_{\text{matching}}$ | $\lambda_{\text{conn}}$ | $k$ |
|---|---|---|---|---|---|---|---|
| BA-2Motif | 0.2 | 2 | 2.0 | 5 | 2.5 | 1 | 5 |
| GroupShapes | 0.1 | 0.6 | - | - | 3.0 | 1 | 5 |
| IMDB-Multi | 0.01 | 0.8 | 7.5 | 17 | 2.0 | 1 | 5 |
| MUTAG | 0.2 | 4 | 8.0 | 6 | 2.5 | 1 | 5 |
| Reddit-Binary | 0.01 | 0.1 | 10.0 | 25 | 2.5 | 1 | 10 |
| OGB-Molhiv | 0.01 | 0.1 | 15.0 | 30 | 15.30 | 1 | 35 |

# F  FURTHER ANALYSIS ON MUTAG

The distinguishing feature of the explanation for the Non-Mutagenic class is the presence of halogen atoms. However, it is important to note that in the literature the presence of halogen atoms is not typically indicative of non-mutagenicity. Upon further examination, we found that matched subgraphs corresponding to this explanation in other Non-Mutagenic instances also contain halogen atoms and achieve high class scores. To investigate this further, we conducted a frequency analysis of halogen atom occurrence across both classes, as shown in Figure 3. The analysis revealed that halogen atoms appear nearly ten times more frequently in non-mutagenic compounds than in mutagenic ones. This suggests that the classifier may have learned to associate the presence of halogens with non-mutagenicity—a pattern that may not hold reliably in real-world scenarios and could lead to misleading predictions.

To validate this hypothesis, we further removed halogen atoms from 10 randomly chosen molecules classified to the Non-Mutagenic class and observed an average score drop of $0.31 \pm 0.009$. This confirms that the classifier indeed relies on halogen presence as a key discriminative feature for predicting non-mutagenicity, even though this association lacks solid chemical justification. Such experiments highlight the risk of spurious correlations embedded in the learned decision process.

It is also worth emphasizing that such insights are made possible due to MATCHEX's motif matching mechanism, which enables systematic investigation of discovered explanations through controlled interventions. In contrast, explanations produced by other methods lack this matching capability, significantly limiting their local fidelity and their ability to support causal probing of model behavior.

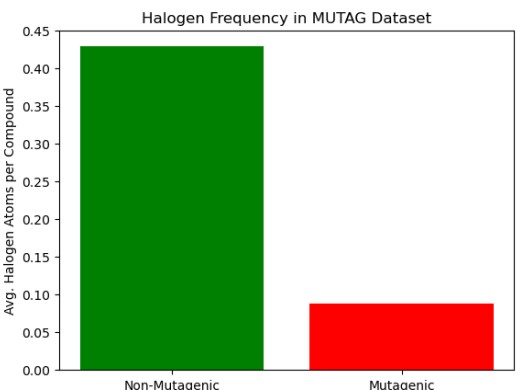

Figure 3: Frequency of halogen atoms across classes in the MUTAG dataset.

# G ALGORITHM OF MATCHEX

Algorithm 2 demonstrates the pseudocode of MatchEx.

---

**Algorithm 2** Implementation of MatchEx to Discover Explanations

---

**Require:** Trained GNN $f$, class-wise graphs $\mathcal{D}_c$, tolerance threshold $\tau$, top-$k$
**Ensure:** Set of model-level or subgroup-level explanations $\mathcal{M}$
 1: $\mathcal{M} \leftarrow \emptyset$
 2: *# Step 1: try single explanation once via MGM on the whole class*
 3: Select top-$k$ graphs from $\mathcal{D}_c$ with highest predicted class-$c$ scores
 4: Run multi-graph matching (MGM) on $\mathcal{D}_c$ to obtain alignments $\{U_i\}$
 5: **for** each $G_i$ in top-$k$ **do**
 6:     Train explainer to predict mask $m_i$ over nodes (optimize Eq. 3)
 7: **end for**
 8: Evaluate generalization scores on $\mathcal{D}_c$ by transferring $m_i$ across graphs using $U_j U_i^\top m_i$
 9: **if** best generalization score $\geq \tau$ **then**
10:     Append the corresponding explanation $(G_i, m_i)$ to $\mathcal{M}$; **return** $\mathcal{M}$
11: **end if**
12: *# Step 2: binary MGMC on the whole class (first split), then refine only failing clusters*
13: $(\mathcal{C}_1, \{U_i^{(1)}\}), (\mathcal{C}_2, \{U_i^{(2)}\}) \leftarrow \text{MGMC}(\mathcal{D}_c, K{=}2)$
14: $\mathcal{Q} \leftarrow [\,(\mathcal{C}_1, \{U_i^{(1)}\}),\ (\mathcal{C}_2, \{U_i^{(2)}\})\,]$
15: **while** $\mathcal{Q}$ not empty **do**
16:     $(\mathcal{C}, \{U_i^{\mathcal{C}}\}) \leftarrow \text{pop}(\mathcal{Q})$
17:     Select top-$k$ graphs from $\mathcal{C}$ with highest predicted class-$c$ scores
18:     **for** each $G_i \in$ top-$k$ of $\mathcal{C}$ **do**
19:         Train explainer to predict mask $m_i$ over nodes (optimize Eq. 3)
20:     **end for**
21:     Evaluate generalization scores within $\mathcal{C}$ using the provided alignments $\{U_i^{\mathcal{C}}\}$ and transfers $U_j^{\mathcal{C}}(U_i^{\mathcal{C}})^\top m_i$
22:     **if** best generalization score $\geq \tau$ **then**
23:         Append the corresponding explanation $(G_i, m_i)$ to $\mathcal{M}$; **continue**
24:     **else**
25:         $(\mathcal{C}_L, \{U_i^{(L)}\}), (\mathcal{C}_R, \{U_i^{(R)}\}) \leftarrow \text{MGMC}(\mathcal{C}, K{=}2)$
26:         push $(\mathcal{C}_L, \{U_i^{(L)}\})$ and $(\mathcal{C}_R, \{U_i^{(R)}\})$ into $\mathcal{Q}$
27:     **end if**
28: **end while**
29: **return** $\mathcal{M}$

---

# H   DETAILED ANALYSIS ON THE OGB-MOLHIV DATASET

## H.1   EXPLANATIONS ON THE MAJORITY CLASS

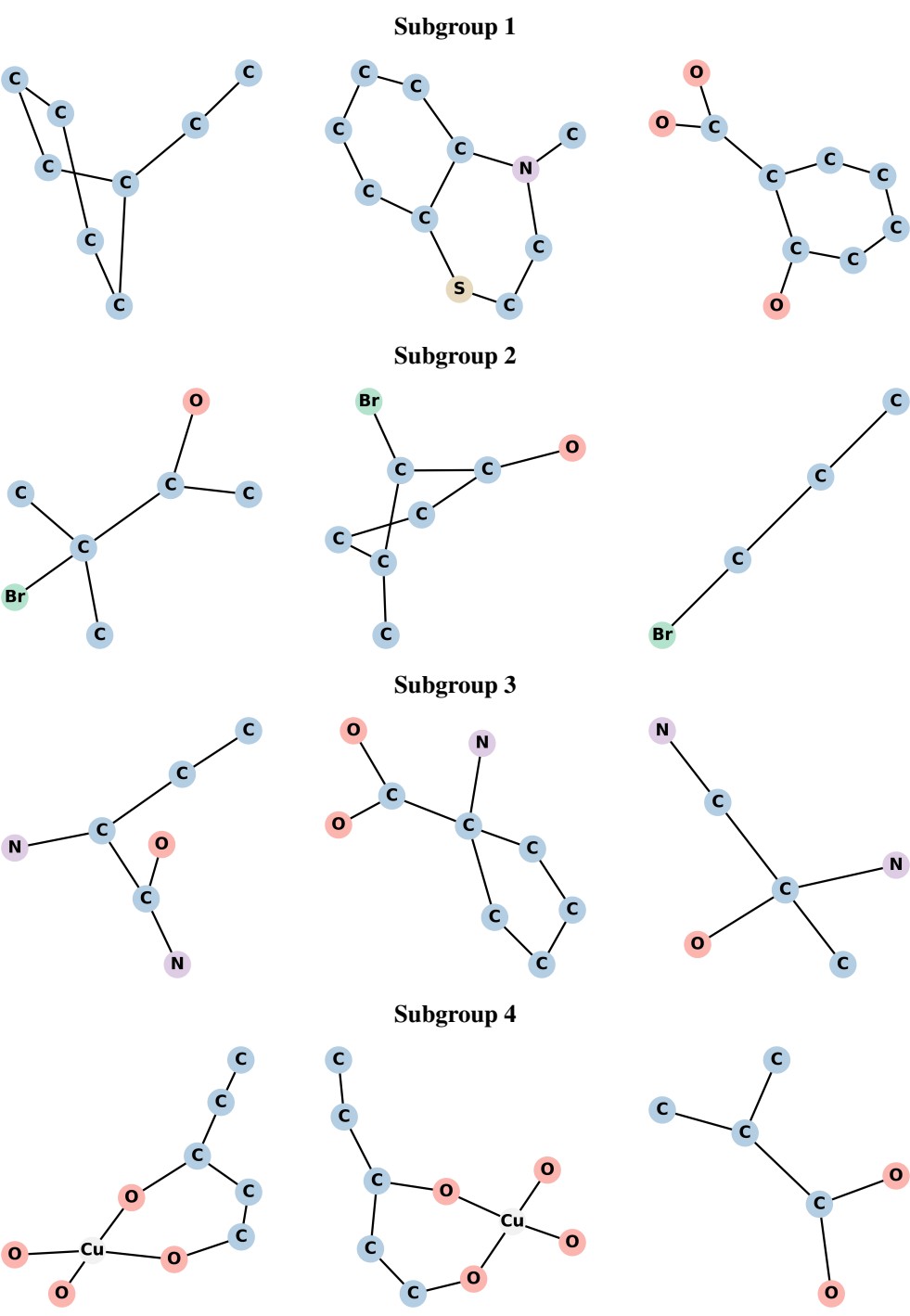

Figure 4: Explanations Corresponding to the Majority Class of the OGB-Molhiv dataset by MatchEx. Each row depicts 3 explanations with highest class score in each subgroup discovered by MatchEx.

In the majority (non-active) class of OGB-Molhiv, MatchEx discovered **four distinct subgroups**, each characterized by a recurrent structural motif. Representative explanations from each subgroup are shown in Figure 4.

- **Subgroup 1 (Aromatic Rings):** Molecules containing benzene-like aromatic rings formed a consistent subgroup. This is chemically plausible, as aromaticity frequently contributes to stability and is overrepresented in drug-like molecules.
- **Subgroup 2 (Halogen Substituents):** Molecules featuring halogen atoms (Cl, Br, F, I) bonded to carbon emerged as a separate subgroup. These motifs often modulate the solubility and bioavailability of the ligand, making them relevant discriminative features.
- **Subgroup 3 (Primary Amines):** Another subgroup was characterized by the presence of $N$ atoms-$NH_2$ groups attached to carbon backbones. Primary amines are common functional groups in pharmaceutical chemistry and were consistently captured by MatchEx.
- **Subgroup 4 (Carbonyl Groups):** Finally, a subgroup enriched with carbonyl groups (C=O) was identified. Carbonyl functionalities are widespread in organic molecules and provide a distinct rationale for classification.

These subgroup-level explanations highlight that the majority class is structurally diverse, and a single motif cannot adequately generalize across all instances. MatchEx adaptively partitions the dataset and identifies motifs that collectively achieve high generalization scores, yielding a richer and more faithful explanation than baseline methods.

## H.2 Explaining Classifier Behaviour on the Minority Class

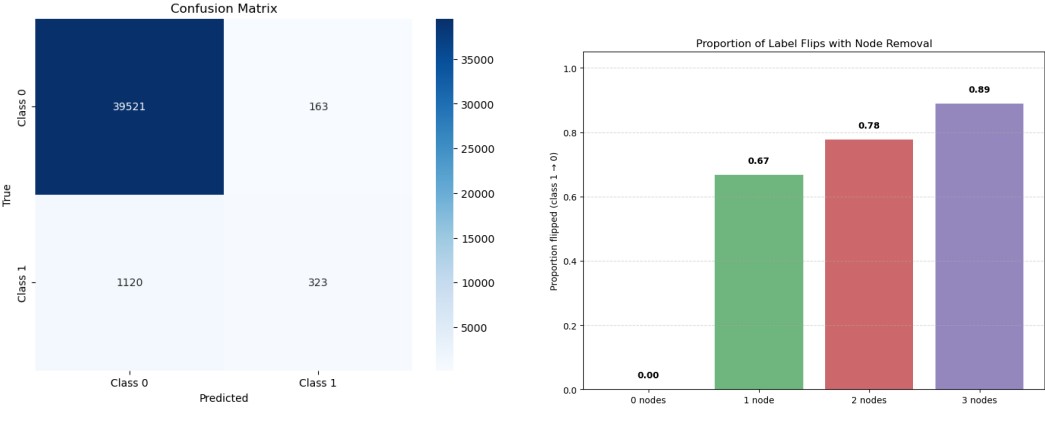

(a) Confusion Matrix of the Classifier.

(b) Proportion of label flipping minority-class graphs

Figure 5: Classifier Analysis on the OGB-Molhiv Dataset

The OGB-Molhiv dataset is heavily imbalanced, with the minority (active) class representing only around 4% of the molecules. Consequently, classifiers trained on this dataset may be inherently biased towards the majority class so much so that it does not learn any general pattern about the minority class. Rather, it only memorizes some instances that it has correctly or wrongly classified to the minority class. To verify if this was indeed the case for the classifier we trained, we conducted a controlled experiment. We picked the top 50 molecules that the classifier had classified in the HIV category by descending order of their class score and progressively deleted one to three nodes from them. Figure 5a shows the confusion matrix of the classifier where it can be seen that the classifier had classified a total of 486 samples ($< 2\%$) of the dataset to the minority class. Out of the 50 graphs we chose from this 486 samples, 67% of their labels flipped to the majority class on deletion of one random node while upto 89% of their labels flipped to the majority class on deletion of 3 random nodes. This confidence drop on samples which the classifier classified to the minority class on deletion of these few random nodes signifies that the classifier has actually memorized these samples rather than learning a general pattern. This explains why MatchEx returned such a high scoring instance as explanation and the generalization score of that instance is near 0.

# I    COMPLEXITY ANALYSIS AND RUNTIME COMPARISONS

We analyze the computational complexity of MATCHEX in comparison with representative model-level explainers. Generative approaches such as XGNN and GNNInterpreter enjoy lower asymptotic complexity, but both face critical limitations in practice. XGNN relies on reinforcement learning to sequentially construct motifs, where the action space of adding nodes and edges grows combinatorially with graph size and node types. This makes policy learning unstable, requiring extensive rollouts and often converging to unrealistic motifs despite high scores. GNNInterpreter adopts a variational sampling approach, but it requires the maximal motif size to be fixed in advance and incurs the full cost of Monte Carlo sampling and optimization for each new explanation, leading to inefficiency and brittle performance on larger datasets. By contrast, discovery-based approaches attempt to identify motifs directly from data. Among them, PAGE performs iterative subgraph search, but the number of candidate subgraphs grows exponentially with graph size, leading to an inherent combinatorial explosion that renders the method unstable on large graphs. MATCHEX avoids such search procedures entirely by formulating explanation discovery as a matching problem, incurring a one-time polynomial cost that amortizes efficiently across all instances of a class while ensuring faithful, multi-granular explanations.

Table 5: Runtime comparison across datasets. For MATCHEX, we report one-time matching and training cost as well as per-explanation sampling time. For XGNN, GNNInterpreter, and PAGE, only per-explanation sampling time is reported.

| Dataset | MatchEx | | XGNN | GNNInterpreter | PAGE |
|---|---|---|---|---|---|
| | Matching and Training Time (seconds) | Sampling Time (seconds) | Sampling Time (seconds) | Sampling Time (seconds) | Sampling Time (seconds) |
| MUTAG | 7.13 | 0.007 | 50.64 | 5.17 | 12.64 |
| IMDB-Multi | 133.35 | 0.004 | 81.31 | 120.43 | 110.31 |
| REDDIT-B | 1834 | 0.004 | 119.63 | 271.85 | 14456 |
| BA-2Motif | 13.39 | 0.006 | 63.32 | 11.12 | 27.71 |
| GroupShapes | 15.61 | 0.006 | 74.49 | 130.23 | 18.92 |
| OGB-MOLHIV | 24228 | 0.363 | – | 178.54 | 40104 |

**MatchEx.**    The complexity of MATCHEX consists of two parts:

$$O(N^2 n^3) + O\big(T(knd^2 + k^2 n^2)\big),$$

where $N$ is the number of graphs in a class, $n$ the average number of nodes per graph, $d$ the embedding dimension, $k$ the number of top graphs retained for alignment, and $T$ the number of explainer iterations. The first term arises from multi-graph matching, which requires solving pairwise alignments with cubic dependence on $n$ and quadratic dependence on $N$. This step is the dominant cost, but it is incurred **once per class**. The second term corresponds to training the explainer, which is lightweight. After alignment, explanations can be generated for all graphs in the class through mask transfer at near-quadratic cost ($O(n^2)$), making the amortized per-explanation cost efficient.

**PAGE.**    PAGE relies on iterative subgraph search to identify motifs. The number of possible subgraphs of a graph with $n$ nodes is $2^{O(n)}$, since every subset of nodes may define a candidate subgraph. Thus, the complexity of PAGE is dominated by

$$O(N_{\max} \cdot 2^n),$$

where $N_{\max}$ is the maximum number of iterations allowed. In practice, PAGE mitigates this by capping $N_{\max}$ and pruning the search, but this introduces heuristic dependence and can easily miss motifs. The key point is that PAGE inherently faces a **combinatorial explosion** in the size of the subgraph search space, which grows exponentially with graph size.

Table 5 reports the mean time across all classes to generate one explanation for all datasets. We report both the matching time and the time taken to sample an explanation. Note, that the matching time is incurred only once per class for MatchEx. The sampling time is incurred each time an explanation is sampled using the corresponding method.

# J    LLM USAGE

We used GPT-5 to polish the writing and grammar of the paper.

