# OpenReview forum: "MatchEx: Model-Level GNN Explanations with Multi-Granular Insights"
_ICLR.cc/2026/Conference — ICLR 2026 Conference Withdrawn Submission_

### Official Review · Reviewer_MwSN · 2025-10-28

**Soundness:** 2
**Presentation:** 2
**Contribution:** 2
**Rating:** 4
**Confidence:** 4

**Summary:**

This paper proposes MatchEx, a framework for “model-level” GNN explanations. Instead of generating a synthetic prototype graph, it tries to recover real subgraph motifs from high-confidence training examples of a given class, align them across multiple graphs by multi-graph matching, and treat those aligned motifs as class or subgroup-level explanations. After reviewing the paper and the appendix, I give a **weak reject rating since several key pieces of the method and its validation are underspecified and there are key flaws.**

**Strengths:**

1. Problem focus is well motivated. Most real users want to know “what pattern defines this class for the model,” not just why one single graph was classified that way.
2. A simple, generally pluggable fix. The pipeline (multi-graph matching with cycle consistency, then enforcing that one learned mask transfers across many graphs) is a neat way to tie together of different level of explanations.

**Weaknesses:**

1. **Scalability / missing details.** MGMC (the “graduated assignment” multi-graph matching plus clustering) is central. The paper says it produces $U_i \in {0,1}^{n_i \times d}$ for all graphs in a class and even partitions huge datasets like OGB-MOLHIV into multiple subgroups. But the main text never explains how $d$ is chosen, how many graphs are jointly matched, how expensive this is, or how it’s made to run at Molhiv scale (~40k molecules). This is a core feasibility question.

2. **Training objective is under-specified.** The loss $L_{\text{MatchEx}}$ mixes a classification term, a “matching alignment” term where the learned mask from one graph is projected into others, a sparsity / budget term, connectivity, and entropy regularizers. It is not clear if $k$ candidate motifs are optimized jointly or independently, what the actual node budget $B$ is, and how the method avoids the trivial “the whole graph is the explanation,” which would obviously keep the class score high and inflate $g_c(M)$. Reproducibility depends on these missing knobs.

3. **Theory over-claims.** Thm 1 argues that maximizing a certain QAP-style objective recovers maximum common induced subgraphs under large $\lambda$. That is fine as intuition for pairwise, near-isomorphic graphs with bounded noise, but the full pipeline uses soft masks, clustering, noisy chemical features, and partial sampling. I think the paper currently phrases this as if it directly justifies MatchEx recovering “true semantic motifs,” which feels stronger than what is actually proven.

4. **Evaluation fairness and clarity.**

   * The “generalization score” $g_c(M)$ is a headline metric, but by design it only applies to real subgraphs that actually occur in the data. Generative baselines like XGNN or GNNInterpreter are not given a $g_c$ score, so MatchEx wins by definition there.
   * Wasserstein-1 distance $W_1$ is measured between motif embeddings and class embeddings after adding Gaussian noise to “densify support.” It is not clear if baselines get the same treatment, or even what embedding space is used.
   * In Table 2 (Seems the author not correctly hightlight some values or metrics, e.g on IMDB-M, this further reduces the soundness of the evaluation results), MatchEx is not always strictly better than GNNExplainer / PGExplainer on instance-level fidelity and infidelity (e.g., on MUTAG and IMDB-M). So the claim that it is the “coherent” solution at all granularities should be toned down.
   * On Molhiv, they report 0.97 accuracy, but Molhiv is extremely imbalanced, so raw accuracy can be dominated by the majority class and the standard evaluation on this dataset should be the AUROC. I would like to see AUROC or F1 for the minority (positive) class and a statement on whether explanations are from train or test graphs.

**Questions:**

1. **MGMC at scale.** How do you actually run MGMC on OGB-MOLHIV? Do you match all graphs at once, or do you subsample top-confidence ones? How is $d$ (the shared node universe size) set? Please give at least rough complexity and memory numbers.

2. **Mask learning details.** Are the $k$ candidate motifs trained jointly in one optimization or one-by-one (each anchored on a different high-confidence graph)? When you transfer a mask $m_i$ to $G_j$ via $U_j U_i^\top$, is the loss symmetric over all pairs or always “anchor → others”? This matters for reproducibility.

3. **Avoiding trivial full-graph explanations.** How exactly do you enforce the “budget” $B$ and sparsity so that the method cannot just keep almost the whole graph and get an easy high score and high $g_c(M)$? Can you report the average fraction of nodes kept in the final motifs?

4. **Metrics on Molhiv.** Since Molhiv is very imbalanced, can you report AUROC / F1 (AUROC is the standard evaluation metric on this dataset) for the minority class and clarify whether subgroup motifs are learned on train, val, or test molecules? Otherwise it is hard to tell if MatchEx is surfacing a real generalization bias or just memorized training artifacts.

---

### Official Review · Reviewer_wFVn · 2025-10-29

**Soundness:** 3
**Presentation:** 2
**Contribution:** 2
**Rating:** 4
**Confidence:** 4

**Summary:**

The proposed MatchEx is a framework that reveals important and recurring subgraph patterns from training graph data by incorporating Multi-Graph Matching and Clustering (MGMC).

**Strengths:**

1. Finding important subgraph patterns from real instances ensures providing more valid model-level explanations, since these patterns actually exist in reality, compared with generative model-level explanations, which risk generating invalid or unrealistic patterns.
2. The discovered important subgraph patterns serve as both global and local explanations. As a result, the local explanations provide more representative performance since they are also optimized from a model-level perspective.

**Weaknesses:**

1. Generally, motifs are defined as "sub-graphs that repeat themselves in a specific network or even among various networks." However, in this paper, "motif" refers more specifically to important and repeated subgraphs. Without providing a clear definition of "motif" in the introduction, using this term may cause confusion by misleading the context and evoking associations with existing methods like MAGE or Motif Explainer. I recommend addressing this terminology issue to avoid confusion.

2. In Theorem 1, the lower bound in the proof for the non-isomorphic case seems too loose. While clustering them into the same cluster might be semantically reasonable, it appears difficult to generalize isomorphism.

3. MAGE and GLGExpaliner, as the relevant baselines, are not compared in the experiments.

**Questions:**

1. Since each graph has a different size, how is the mask applied across the top k graphs?
2. In Equation (3), $f_{c}$ is described as the class score. Does this mean you are minimizing the class score in both $L_{cls}$ and $L_{matching}$? Is a higher $m$ value indicative of more important nodes, or vice versa? Meanwhile, sparsity and budget should be minimized. Please clarify the final training objective function and whether you are maximizing or minimizing each component.
3. The authors mention using a tolerance threshold of 0.7 in lines 207-208. Isn't this too generous compared to the standard 0.5 or the threshold less than 0.5? For example, this assumes that a drop in class score from 0.9 to 0.2 (0.9 - 0.7 = 0.2) would still be considered consistent.
4. Please refer to the weaknesses mentioned above.

---

### Official Review · Reviewer_bzZc · 2025-10-30

**Soundness:** 2
**Presentation:** 2
**Contribution:** 2
**Rating:** 2
**Confidence:** 4

**Summary:**

The paper introduces MatchEx, a model-level explainability framework for Graph Neural Networks. Their motivation is that existing generation-based models have the following shortcomings:
1. Generated motifs often fail to resemble real substructures in the data.
2. They capture only a narrow range of class-specific motifs and cannot reflect diversity.
3. They lack mechanisms to connect model-level explanations to instance-level reasoning.

Therefore they favor motif-extraction-based approaches, and propose MatchEx, which formulates model-level explanation as a multi-graph matching and clustering (MGMC) problem. They introduce a generalization score to assess whether one motif generalizes across all class instances. If not, MatchEx adaptively partitions graphs into coherent subgroups and extracts subgroup-specific motifs. The explainer is trained under an information bottleneck-inspired objective, balancing interpretability (high class score) and conciseness (small subgraph).

**Strengths:**

1. Proposes the first matching-based model-level explainer for GNNs.
2. Bridges global, subgroup, and instance-level explainability within one unified framework.
3. Introduces a generalization metric that also serves as a diagnostic tool for detecting model bias.
4. Demonstrates state-of-the-art performance across six diverse benchmarks.

**Weaknesses:**

1. Important related works are missing, such as TreeX and GCNeuron. In fact, TreeX focuses on exactly the same problem as MatchEx, bridging global- and instance-level explainability, capturing class-specific subgraph-based explanations, and might be more efficient than MatchEx. But the authors did not even mention it.
2. Overemphasis on generative explainers and incomplete framing of the field.
Generation-based methods are only one branch of model-level GNN explainability. The introduction leans heavily on criticizing generative approaches (e.g., XGNN, GNNInterpreter), giving the impression they constitute most prior work. In fact, there is a parallel line of motif-extraction/discovery methods (e.g., MAGE, GLGExplainer, TreeX) targeting a similar goal to MatchEx. However, they are not integrated into the paper’s narrative as co-equal alternatives and are not compared in experiments. The paper itself claims many existing model-level methods are generative and details their drawbacks, which skews the framing.
3. Comparisons misaligned with the method family. The experiments compare MatchEx only with three model-level baselines (XGNN, GNNInterpreter, and PAGE), all of which the paper itself labels as generative or search-based.
But since MatchEx is a motif-extraction / matching-based approach, these baselines are not the most appropriate. Fair evaluation would require including other motif-extraction model-level explainers (e.g., MAGE, GLGExplainer, TreeX) that tackle similar objectives of discovering semantically consistent motifs across instances.
4. For instance-level evaluation, only GNNExplainer (2019) and PGExplainer (2020) are compared. These are early local explainers and no state-of-the-art instance-level baselines (e.g., GraphMask, SubgraphX, CF-GNNExplainer, or more recent 2023–2024 methods) are considered.

**Questions:**

N/A

---

### Official Review · Reviewer_3N5B · 2025-11-01

**Soundness:** 2
**Presentation:** 3
**Contribution:** 2
**Rating:** 4
**Confidence:** 3

**Summary:**

The paper proposes MatchEx, a discovery based framework for model level explanations of GNNs that (i) aligns semantically similar, not-necessarily isomorphic motifs across graphs using a multi graph matching and clustering (MGMC) objective (graduated assignment), (ii) selects a class level motif by maximizing a generalization score that measures how well the motif transfers to other instances via learned matchings, and (iii) projects the global rationale back to instance level explanations through mask transfer. MatchEx outperforms XGNN, GNNInterpreter, and PAGE on model level metrics (target class score, generalization score, Wasserstein 1), and yields competitive instance level fidelities.

**Strengths:**

S1. The core idea—optimize a matching objective that recovers semantically similar motifs and then use those alignments to both select a model level rationale and project it to instances—is well motivated and addresses a real gap between exact motif search and generative explainers.

S2. Across datasets, MatchEx typically yields higher performance. MatchEx surfaces distinct subgroup motifs where baselines collapse to one explanation.

S3. The paper provides architectural details and an (anonymous) code link, enabling verification.

**Weaknesses:**

W1.
The one time complexity O(N^2 n^3) and the 24,228 s matching time on OGB MOLHIV limit immediate use on very large classes. Even if amortized, this is heavy. Authors may add experiments with approximate matchers (e.g., GW style relaxations or learned matchers) and report wall clock/memory scaling vs. N,k,n.

W2.
Eq. (4) measures score differences within \gamma=0.1; this is sensitive to model calibration and class imbalance, and \gamma is fixed across datasets. Could you report sensitivity to \gamma and alternative thresholds (probability vs. logit)?

W3. The method hinges on U_i ∈ {0,1}^{n_i×d} but the selection of d (and its impact on stability/quality) is not detailed.

W5. Comparisons focus on XGNN/GNNInterpreter (generative) and PAGE (search). Contemporary non generative model level methods such as GLGExplainer and MAGE are not evaluated, though they are conceptually close.

W6. How are motif/graph embeddings computed for the W1 metric—penultimate layer graph embeddings from the same classifier, or a fixed encoder?

**Questions:**

Please refer to the weaknesses above.

---

### Author Response · Authors · 2025-12-02
**Official Comment by Authors in Response to Reviewers' Critiques**

We thank the reviewers for their time. We would like to address some aspects of the critiques below in a single common answer:

> **Why not compare with MAGE and GLGExplainer?**

We have already mentioned in the paper that MAGE is only applicable to molecular graphs. The authors also mention this in response to a reviewer's comment on their OpenReview page (see Q5 from Reviewer KGJQ). Therefore, it is not applicable to most datasets we use in the paper. GLGExplainer has a completely different goal. The authors try to build model-level explanations using concepts extracted from training instances which is very different from what we try to achieve.  The goal of MatchEx is :

 a) **Identifying Subgroups of Instances in a Target Class that Share a Common Explanation for belonging to the target class**

b)**Discover the group-level explanation for each identified subgroups**

c) **Show a matching explanation in each instance of the group showing why the group-level explanation makes sense**.

Therefore unlike what the reviewers mention, the mentioned methods are not any more related to our work than the model-level explanation techniques we already compare with.

> Why is the method computationally heavy and why not use approximate matchers to reduce time complexity?

We agree that our method is computationally heavy on large datasets and we have explicitly mentioned it as a limitation. However, the computational cost should not be seen in isolation. It should be seen in relation to what we stand to gain by paying this cost. To the best of our knowledge, this is the first method that ties together the concept of model and instance level explanations. As Reviewer MwSN mentions, matching provides a neat way to directly establish a relationship between explanations at various granularities. **Establishing this relationship is the central contribution of the paper. It provides a way to answer why is the explanation for a class $c$ relevant to instance X that the classifier has categorized to class $c$.** Secondly, the method does not require a secondary instance-level explanation method unlike GLGExplainer to output this instance-level explanation. The matching mechanism is enough in itself to provide a explanation for each instance that directly relates it to the common model-level explanation. Finally, as we discuss in the paper extensively **cycle consistency** is a unique property that the GA-GM algorithm provides and is important in our context. Approximate matchers do not provide this guarantee and as such using such a method would mean throwing out our method and writing another paper altogether.

> On different ablations that the reviewers' suggested on universe dimension $d$ and tolerance threshold $\gamma$

We agree that ablations are important for demonstrating stable performance. However, we would like to remind the reviewers that we have demonstrated already through ablations that the performance of our method remains stable with variation in entropy and sparsity constraints. We have also demonstrated that the generalization capability of the explanations increase when the matching alignment term is added. The suggested ablation on universe dimension $d$ and $\gamma$ would not add any new insight in our view. $d$ is an expressivity parameter and we found that setting it to anything greater than the dimensionality of node feature vectors gives reasonably good performance. On the tolerance threshold,  it is easy to estimate the effects of varying it.  Increasing it would increase the number of subgroups discovered which would only inflate the performance of the method because the subgroups would be smaller and MatchEx would have an easier task of finding common explanations. Decreasing it, would cause no effect on datasets where the target class is one group, it might decrease the generalization score somewhat on classes which were clustered into different groups. In any case, we do not have the computational resources to conduct a thorough ablation on these many parameters on large datasets, hence we would not be able to comply with the reviewers suggestion

> Why not compare with recent instance-level baselines to show we beat them?

Beating instance-level baselines is not what our contribution is about. The comparison we conduct with GNNExplainer and PGExplainer is to show that the matching explanations to model-level explanations are not non-sensical. They make sense as instance-level explanations as seen by common instance-level metrics. We do not try to propose a SOTA instance-level method. We compare our method with recent model-level methods since MatchEx is a model-level method and we beat the compared methods on diverse metrics comprehensively.

---

> ### Author Response · Authors · 2025-12-03
> **Continuation of the Above Comment**
>
> In summary, we do not agree with the major concerns raised by the reviewers about this paper.  We feel that they are not representative of the content of this paper. Since the option for the reviewers to reply to our comments also closed before we could have a discussion regarding our contribution with the reviewers, we have decided to withdraw this paper from this venue.  We thank the reviewers again for their time and efforts.

---

### Note · Authors · 2025-12-03

I have read and agree with the venue's withdrawal policy on behalf of myself and my co-authors.